# Factors Predicting CT Pulmonary Angiography Results in the Emergency Department

**DOI:** 10.3390/diagnostics15070827

**Published:** 2025-03-25

**Authors:** Nika Rakuša, Zrinka Sertić, Maja Prutki, Ana Marija Alduk, Ivan Gornik

**Affiliations:** 1School of Medicine, University of Zagreb, 10000 Zagreb, Croatia; nika.rakusa1@gmail.com (N.R.); maja.prutki@gmail.com (M.P.); aalduk@gmail.com (A.M.A.); 2Department of Internal Medicine, University Hospital Centre Zagreb, 10000 Zagreb, Croatia; zrinka.sertic1@gmail.com; 3Department of Diagnostic and Interventional Radiology, University Hospital Centre Zagreb, 10000 Zagreb, Croatia; 4Department of Emergency Medicine, University Hospital Centre Zagreb, 10000 Zagreb, Croatia

**Keywords:** pulmonary embolism, CTPA yield, predictive factors, D-dimer, heart failure, COPD exacerbation

## Abstract

**Background**: Pulmonary embolism (PE) remains a major concern in emergency patients presenting with respiratory symptoms, with an increase in the demand for CT pulmonary angiography (CTPA) and low yields of this ever more sensitive test. We wanted to investigate factors associated with pulmonary embolism on CTPA, aiming to reduce unnecessary requests. **Methods**: In a single-center, retrospective study, we analyzed all CTPA reports for emergency patients during the year 2023. Various patients’ variables were evaluated for associations with the presence/absence of PE, including the presence or absence of pulmonary pathology identified prior to the CTPA order. **Results**: A total of 1555 CTPA reports were analyzed, of which 278 (17.9%) were positive for PE. The highest ORs (40.9) for PE were found for patients diagnosed with DVT prior to CTPA. The lowest odds ratios of having PE were found for patients with acute congestive heart failure (OR = 0.141), especially in the absence of cancer (OR = 0.089) and for patients with hypercapnia in COPD exacerbation (OR = 0.062). Tachycardia and hypoxemia were the physiological variables positively associated with PE, while hypercapnia was negatively associated with PE. For patients with heart failure, COPD exacerbation, and pneumonia, higher D-dimer cut-off values (3.87 mg/L, 1.25 mg/L, and 1.34 mg/L, respectively) were found to retain 100% sensitivity for PE. **Conclusions**: Stricter criteria for CTPA orders in the presence of other pulmonary pathologies may reduce unnecessary scanning. Higher D-dimer cut-off values in such cases may lead to higher specificity without sacrificing sensitivity.

## 1. Introduction

Venous thromboembolism (VTE), presenting as deep venous thrombosis (DVT) and pulmonary embolism (PE), is a significant cause of morbidity and mortality worldwide [1]. The incidence of PE in the Western world is reported to be ~1 per 1000 person-years but varies widely when stratified by demographics, risk factors, and comorbidities [2,3].

The incidence of PE has steadily increased over the past few decades [2,4,5], which can be attributed to several factors. In developed countries, there is a demographic trend of aging populations with a sedentary lifestyle and more chronic health conditions, which are all increasing the risk for VTE. Moreover, advancements in imaging techniques allow for higher sensitivity and the detection of smaller emboli [6]. This can lead to ambiguous findings being reported as positive, resulting in an increase in the number of reports with only subsegmental and small pulmonary artery contrast defects which have low interobserver agreement [7] and questionable clinical relevance. Additionally, during the SARS-CoV-2 pandemic, which was a global health crisis, an increased risk of thrombosis was reported in hospitalized patients [8]. Heightened awareness of VTE among healthcare providers when exposed to the sum of these data has led to a substantial rise in the utilization of emergency department (ED) resources for investigating PE. In our institution, a comparison of the CT pulmonary angiography (CTPA) usage in the ED before and during the SARS-CoV-2 pandemic revealed a 3.4-fold increase in volume, while the yield and rates of significant PE (involving the pulmonary trunk and main or lobar arteries) were not substantially different, indicating an increase in the absolute number of negative scans [9]. Similar trends are observed among other published literature [10,11].

The diagnostic algorithm for suspected pulmonary embolism [12,13,14] generally includes clinical assessment and pre-test probability assessment using the Geneva score [15] or Wells score [16] with D-dimer testing to rule out PE in low-risk patients. Patients with low pre-test probability and elevated D-dimers and those with high pre-test probability are further investigated with CTPA to confirm or exclude the diagnosis. Ventilation-perfusion lung scintigraphy (V/Q scan) is an alternative if CTPA is not available or contraindicated.

There is a delicate balance between the risk of missed or delayed diagnoses, which can lead to increased morbidity and mortality, and the risk of over-investigation with advanced imaging methods, which can result in misdiagnosis, overtreatment of insignificant findings, and prolonged times in the ED. No less important is the risk of contrast-enhanced CT scans, which not only expose patients to relatively high levels of radiation but may also acutely cause contrast-induced nephropathy. All mentioned risks present an important peril to patients and healthcare systems overall. Conditions such as malignancy, heart failure, chronic pulmonary disease, and acute respiratory infections present similarly but have different pre-test probabilities of VTE. Identifying patients with low or negligible likelihood of PE is essential to enhance individual outcomes and improve efficiency in the ED.

To identify patient profiles associated with higher or lower odds of PE diagnosis, we conducted a retrospective, single-center review of patients who underwent CTPA for suspected PE at a tertiary care academic hospital’s ED. We also collected data on patients presenting with respiratory symptoms during the study period, assessing their leading symptoms at admission and diagnosis at discharge to better understand the context of the patients on whom CTPA scans were performed.

## 2. Materials and Methods

This retrospective study was conducted at University Hospital Centre Zagreb, Zagreb, Croatia, and included adult patients (18 years or older) admitted to the ED presenting with respiratory symptoms from 1 January 1 to 31 December 2023. In all patients, we reviewed electronic medical records for symptoms at admission and final diagnosis at discharge. In patients who underwent CTPA for suspected PE, we collected the following data: demographic information, such as sex and age at presentation; presenting symptoms including dyspnea, cough, hemoptysis, chest pain, or fever; physiological parameters, including pulse rate, oxygen saturation, blood pressure, PaCO_2_; the presence of active malignancy, defined as any cancer diagnosed or treated within 6 months of index presentation; whether the patient was on oral anticoagulant treatment at the time of index presentation; and history of SARS-CoV-2 infection during or within two weeks prior to presentation.

Ordering CTPA scans in our center is based on the Wells PE score [16] and D-dimer determination in accordance with NICE guidance [17]. The CTPA scans were interpreted by the attending radiologist at the time of admission, and the original, unrevised radiology reports were collected to establish the presence or absence of PE on CTPA. The D-dimer concentration was determined in the local laboratory using a Siemens Innovance^®^ (Siemens Healthineers AG, Erlangen, Germany) D-dimer immunoturbidimetric assay with calibrators and controls from the same manufacturer.

We also analyzed diagnostic information acquired before the CTPA request. This included evidence of acute COPD exacerbation (eCOPD) based on standard criteria [18,19]; evidence of acute congestive heart failure, defined as pulmonary congestion observed via either ultrasound or X-ray along with elevated NT-pro BNP values according to age-corrected cut-offs [20]; a diagnosis of pneumonia confirmed on chest X-ray accompanied by elevated inflammatory markers; and the presence of deep vein thrombosis confirmed by compression ultrasound or a CT scan.

Results were analyzed with MedCalc v22.017 statistical software. Data are presented as absolute and relative frequencies or arithmetic means with standard deviations as appropriate. To convey the association of PE with specific risk factors and comorbidities, an odds ratio with a 95% CI was calculated. Comparisons of continuous variables were performed using Student’s *t*-tests. Logistic regression was performed with the “Enter” method with variables included if *p* < 0.05 and variables removed if *p* > 0.1. ROC curves were constructed using the method of DeLong et al. [21] with a binomial exact confidence interval for the AUC. All *p*-values are two-tailed, and statistical significance was set at an α level of 0.05.

## 3. Results

During the investigated period, 131,945 patients were admitted to the ED, of which 13,366 presented with respiratory symptoms: 5957 (44.6%) had dyspnea, 5341 (40.0%) had chest pain, 1545 (11.6%) had a cough, and 523 (3.9%) had hemoptysis recorded as the leading symptoms. Among these, the primary diagnosis was identified as acute congestive heart failure in 2158 (16.1%) patients, pneumonia in 1782 (13.3%) patients, eCOPD in 1480 (11.1%) patients, acute coronary syndrome in 1543 (11.5%) patients, and pleural effusion in 514 (3.8%) patients.

A total of 1555 CTPAs were performed during the analyzed period, of which 278 (17.9%) were positive for PE. Adherence to local CTPA ordering guidelines was good, since only 4.3% of patients were scanned with low Wells scores and D-dimer levels lower than the age-adjusted threshold. However, 26 patients (1.7%) had D-dimer levels lower than the non-adjusted threshold (0.5 mg/L).

Among patients who underwent CTPA, 599 (38.5%) had at least one possible cause of symptoms identified before ordering CTPA: 305 (19.6%) had pneumonia, 205 (13.2%) had acute congestive heart failure, and 96 (6.2%) had eCOPD. Fifty-one patients (3.3%) were sent to CTPA after establishing DVT.

Patients with PE were older than those without PE, but the difference was not statistically significant in our cohort (69.2 ± 14.6 vs. 67.3 ± 15.7, respectively; *p* = 0.066). Pulse rates at presentation were significantly higher in patients with PE compared to those without PE on CTPA (98.5 ± 23.9 vs. 92.6 ± 21.6, respectively; *p* = 0.021). Hemoglobin oxygen saturation was significantly lower in patients with PE compared to those without PE (92.3 ± 7.6% vs. 92.9 ± 6.9%, respectively; *p* = 0.023). Partial pressures of CO_2_ in arterial blood were significantly lower in patients diagnosed with PE than in those without (4.7 ± 1.1 vs. 5.2 ± 1.9, respectively; *p* = 0.002).

Factors related to increased or decreased odds of having PE on CTPA are shown in Table 1. The most significant factor associated with PE was the presence of DVT, which was associated with a diagnosis of PE in 88% of the patients (OR: 40.91; 95% CI: 17.26–96.99). Patients with active cancer, especially those without an obvious cause for their symptoms, exhibited significantly higher odds of having PE. Oral anticoagulant treatment did not affect the probability of a patient having PE in the cancer population or in the total cohort.

Patients without an identified cause of symptoms (specifically pneumonia, eCOPD, or AcHF) prior to the CTPA request, regardless of any other factors, also had significantly higher odds of PE. This was particularly evident in the subgroup of patients older than 65 years, and even more so in those aged 75 or older, who had almost a two-fold increase in the probability of having PE if there was no identified cause for their symptoms.

Among the symptoms and signs, tachycardia, dyspnea, and chest pain were found to be associated with increased odds of having PE. Furthermore, dyspnea and tachycardia that could not be explained by other causes prior to the CTPA order had even higher ORs for PE. In contrast, hypercapnia was associated with two-fold lower odds of having PE.

Patients diagnosed with acute congestive heart failure prior to the CTPA order had significantly lower odds of PE, especially those without cancer (nine-fold lower OR) or those on oral anticoagulant therapy (eleven-fold lower OR). There were no patients with heart failure without cancer and on oral anticoagulant therapy that simultaneously had PE, although this subgroup was rather small (*N* = 37). Similarly to heart failure patients, patients with eCOPD had very low odds of having PE, especially those with hypercapnia, none of whom was found to have PE in our study.

When considering all patients (*N* = 2363) diagnosed with heart failure during the investigated period (205 who underwent CTPA and 2158 in whom PE was excluded without CTPA imaging), only seven patients (0.3%) were diagnosed with PE. Equally, the likelihood of an eCOPD patient having PE, when considering all 1878 eCOPD patients (96 who underwent CTPA and 1782 in whom PE was excluded without CTPA) during the analyzed period, is 0.27%, as only 5 patients with eCOPD had PE.

A multivariate logistic regression was performed to identify factors independently associated with PE on CTPA. The analysis identified that acute heart failure, eCOPD, and pneumonia were independently associated with the absence of PE, and dyspnea, tachycardia, and the D-dimer value were independent predictors of a CTPA scan positive for PE (Table 2).

Additionally, we examined the value of the D-dimer testing in patients with established heart failure and suspected PE. ROC curve analysis showed that the area under the curve (AUC) was greater for D-dimers in patients with heart failure compared to the overall population undergoing CTPA: 0.900 vs. 0.768, respectively (Figure 1). Analysis of the ROC curve for acute heart failure patients showed that a D-dimer value exceeding 3.67 had 100% sensitivity and 75.6% specificity for PE, with a negative likelihood ratio of 0.00. The same investigation was conducted for other diagnoses established before CTPA orders, and the results are presented in Table 3. Areas under ROC curves were similar in all conditions, and higher cut-off values for D-dimers retained 100% sensitivity with substantial gains in specificity.

## 4. Discussion

In our study, the CTPA diagnostic yield of 17.9% was high compared to some reports from the USA, where the yield was as low as ~2–3% [22,23]. However, positive CTPA rates vary widely across regions and healthcare systems, making it challenging to define an acceptable yield range. Reports from Australasia found yield rates of 12.4% and 14.6% [24,25], which are similar to our findings but lower than the European average found in a meta-analysis comparing European and American yields, where European studies reported an average diagnostic yield rate of 30%, more than twice as high as the 13% pooled from studies from North America [26]. Our yield rate certainly falls within the middle range, and there may be potential for further improvement through closer scrutiny of specific subpopulations. Adherence to guidelines for CTPA requests was mostly maintained, but our results show that in many cases, the unadjusted D-dimer cut-off value (0.5 mL/L) was applied to patients with low Wells scores. Applying age-adjusted cut-offs would have prevented 42 CTPA scans, all of which were negative for PE.

We demonstrated that the odds of positive CTPA scans varied widely across different patient subgroups for whom CTPA was ordered in the ED. The highest ORs were found in patients with DVT, while the lowest were observed in those with acute congestive heart failure, particularly among patients on oral anticoagulants, and those experiencing eCOPD, especially if they presented with hypercapnia. Conversely, patients with symptoms unexplained by other causes were roughly twice as likely to have a positive scan.

Since nearly 90% of patients with established DVT in our cohort had PE, our data support current guideline recommendations [12] that CTPA is unnecessary for confirmed DVT cases with symptoms suggestive of PE, which, in those cases, is assumed. In such patients, further testing should instead focus on assessing the PE severity and early mortality risk. Treatment of DVT and of mild and moderately severe cases of PE is the same (anticoagulation), rendering confirmation of PE diagnosis unnecessary even from the practical standpoint. The diagnosis should nevertheless be confirmed in unstable or high-risk patients since they are or could be candidates for high-risk treatment, namely, thrombolysis or thrombectomy.

Blood stasis, hypercoagulability, and endothelial dysfunction (Virchow’s triad) predispose heart failure patients to an increased risk of thrombosis [27]. However, we took a special interest in patients presenting to the ED with respiratory symptoms in whom diagnostic workup confirmed acute congestive heart failure, motivated by the assumption that the hemodynamics of acute congestive heart failure and PE contradict one another. Specifically, embolism to pulmonary arteries should lower the pulmonary capillary pressure and left-ventricular filling pressures [28], which are the primary driving forces of lung congestion in acute heart failure, making their coexistence improbable. Accordingly, our results clearly indicate that the likelihood of a patient presenting to the ED with acute congestive heart failure and also having coexisting PE is negligible (< 0.5%). Even among patients for whom concurrent circumstances raised suspicion of PE and who underwent CTPA, the probability of PE remained very low, especially in patients without malignant disease, those on anticoagulant treatment, or both. The conditions leading to CTPA orders in patients with acute heart failure cannot be objectively assessed with our study design but are certainly multifactorial. The data nonetheless show that this subgroup of patients could be a good target for reducing the number of unnecessary CTPA orders.

Another subgroup of special interest were patients with eCOPD. On the one hand, in these individuals, the pathophysiological conditions, namely, bronchiolar and bronchial obstruction, primarily lead to alveolar hypoventilation and hypercapnia, with more severe cases also experiencing hypoxemia. On the other hand, PE causes hyperventilation in the areas of the lung unaffected by emboli, which makes the blood coming from these areas hypocapnic. The mixing of blood from both affected and unaffected regions results in normocapnia or even hypocapnia, depending on the ratio of affected to unaffected areas. Therefore, hypercapnia in PE should not be expected unless there is a significant obstruction of pulmonary circulation, typically associated with circulatory compromise (obstructive shock) [28]. Such patients are much less common and would and should in any case follow a separate diagnostic routine including CTPA. In our dataset, none of the hypercapnic patients had PE, and the proportion of PE patients was very low within the overall eCOPD group. However, these results should be interpreted with caution, as the low probability of PE in our eCOPD population is in discordance with other reports. Two meta-analyses found pooled prevalences of 12% and 17.6 in patients with eCOPD [29,30]. Another study raised concerns about the high rates of PE in otherwise unexplained eCOPD [31]. It should be noted that the results varied significantly among studies and were lower in the ED setting compared to the inpatient setting. Also, lack of hypercapnia was consistently associated with PE. It is also possible that CTPA was omitted in the ED and a delayed diagnosis of PE was established later during the in-hospital stay in patients with more severe symptoms.

As it is common knowledge that patients with cancer are in a hypercoagulable state, it is of no surprise that we found a clear association between PE and active malignancy. However, we did not find an increased probability of PE in patients with SARS-CoV-2 infection. The reports on this issue are contradictory [32,33] and may depend on the subtype of the virus that was dominant in the studied populations. Our data suggest that COVID-19 infection should not be viewed as increasing the probability of PE, and the usual criteria should be applied.

The highest percentage of CTPA scans positive for PE was found in patients with hypoxemia (SpO_2_ < 94%) and no identifiable cause in the pre-CT workup. A slightly lower, but still very high, chance of PE was found in patients without an obvious cause of tachycardia (pulse rate > 100/min) or chest pain. These patients should be considered the best candidates for CTPA, which should only be omitted for strong opposing reasons.

Establishing a clear diagnosis in patients who present with respiratory symptoms before considering CTPA may render the scan unnecessary unless there are additional circumstances. A potential objective factor for identifying patients for whom CTPA may still be warranted after establishing another diagnosis that explains the symptoms could be elevated D-dimer levels. Our data suggest that higher D-dimer cut-off values could serve this purpose. D-dimer cut-off values are already adjusted for age in outpatients over 50 years [34], and according to the YEARS study, a D-dimer level adjusted to the pre-test probability assessment retains nearly 100% negative predictive value for the test [35,36,37]. Similar reasoning could suggest higher cut-off values for PE exclusion in cases of other established pulmonary pathologies. Further research in this area could more accurately specify the exact cut-off value for D-dimers in such situations.

Another approach to planning further research would be to integrate previously identified pulmonary pathologies into a pre-test probability score by assigning negative points. An example is the Wells score for deep vein thrombosis [16], where an alternative diagnosis that is equally or more likely than DVT carries two negative points, which significantly lowers the score.

Our data suggest that when a possible cause of respiratory symptoms is identified, additional scrutiny before ordering CTPA is crucial. This should take into account factors such as heart rate, pCO2 levels in the arterial blood, the presence of active malignancy, and anticoagulant therapy. Given the complex nature of these investigations, machine learning studies may offer additional value in the future, especially for specific patient subgroups wherein the existing diagnostic algorithms have limited applicability, such as cancer patients and post-operative patients [38,39].

CTPA carries several potential risks that should be considered before proceeding with the scan. The use of ionizing radiation during CTPA contributes to cumulative radiation exposure, which may increase the risk of developing certain malignancies over time, particularly in individuals who undergo repeated imaging [40,41]. Another significant concern is contrast-induced nephropathy (CIN), a form of acute kidney injury that can occur in patients receiving the iodine-based contrast agent, especially those with pre-existing kidney dysfunction, diabetes, dehydration, hypertension under treatment, hypotension, coronary artery disease, history of nephrotoxic drug use, liver disease, congestive heart failure, age 75 or older, and anemia. The reported incidence of CIN following CTPA is 10–14% in the overall population [42,43]. CIN is associated with an increased risk of severe outcomes, including severe renal failure and death, observed in 12–16% of patients who developed CIN after CTPA. Additionally, CTPA may yield incidental findings, such as benign lung nodules or other unexpected abnormalities, which could lead to unnecessary follow-up testing, anxiety, and invasive procedures that may not have been needed. These risks must be weighed against the diagnostic benefits of CTPA.

Some results may seem surprising, such as the low association of hemoptysis with PE. This can be explained by our hospital’s special role as a national lung cancer center, which increases the number of hemoptysis patients in the ED’s case mix. Other patients’ characteristics and their association with positive or negative CTPA scans are certainly also influenced by an extraordinary case mix.

In this study, we did not include data on family history of VTE, even though evidence suggests that it is a good risk indicator [44]. These data are not routinely collected for patients with dyspnea in our ED, as it is not part of the pre-test risk assessment system we use and was therefore not available due to the retrospective nature of the study. Future investigations should certainly include this information to evaluate it against other risk indicators in a wider context.

Several studies have addressed the issue of CTPA yields for PE [45,46,47,48,49]. Most of them investigated a similar number of CT examinations but primarily focused on yields and the association of basic patient characteristics with the finding of PE. To the best of our knowledge, no study has investigated in depth the characteristics of patients, including physiological data, particularly concerning the correlation between acute congestive heart failure or eCOPD and the likelihood of PE.

In addition to the very detailed acquisition of patient data, the strengths of this study include its broad scope, large patient population, and clear diagnostic criteria. However, we acknowledge several limitations: those inherent to the retrospective nature of data collection and the single-center setting, the absence of preset criteria for PE on CTPA, and the lack of a central radiology review, as inter-observer agreement on the diagnosis of smaller pulmonary emboli may vary [50]. Additionally, the small number of participants in certain patient subgroups, such as those with heart failure without cancer and those on oral anticoagulants, limits the statistical power. The previously mentioned case-mix concerns also limit the generalization of the results. We also recognize that in some patients, clinical judgment may have incorrectly excluded PE without imaging, and the current scope of our study did not include patients who were admitted and for whom the diagnosis of PE was delayed until later during their hospital stay.

## 5. Conclusions

We can conclude that tightening the criteria for CTPA orders when other pulmonary pathologies may explain respiratory symptoms could improve CTPA yields, particularly in cases of eCOPD with hypercapnia and acute congestive heart failure without cancer and/or oral anticoagulant therapy. Our results also support waiving CTPA orders for diagnosing PE in cases of DVT with concurrent symptoms suggestive of PE.

CTPA should, however, be strongly advised in patients with respiratory symptoms coupled with hypoxemia and/or tachycardia that cannot be explained with first-line diagnostic methods (clinical and laboratory testing and chest X-ray or ultrasound).

Larger studies and a prospective setting examining higher D-dimer cut-offs and clinical criteria for establishing suspicion of PE that warrants advanced imaging when other comorbidities are present may lead to more precise recommendations.

## Figures and Tables

**Figure 1 diagnostics-15-00827-f001:**
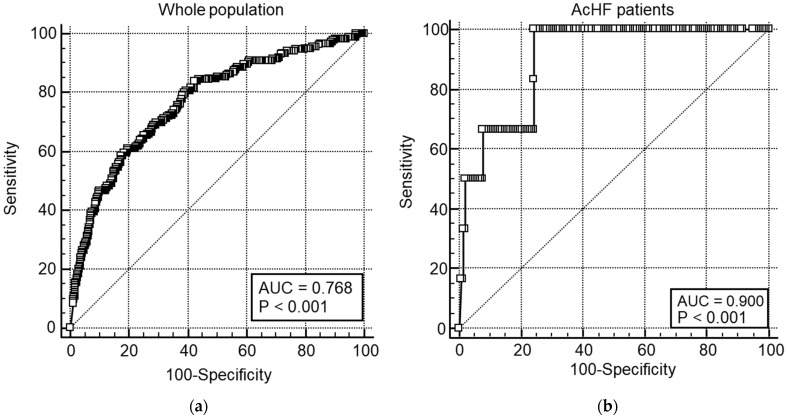
Comparison of ROC curves for D-dimer diagnostic value in (**a**) unselected patients (whole population) and (**b**) acute congestive heart failure (AcHF) patients.

**Table 1 diagnostics-15-00827-t001:** Association of different factors with pulmonary embolism in the population undergoing CT pulmonary angiography. Number of patients with specific characteristic is given in brackets.

Characteristic	With PE	Without PE	OR (95% CI)	*p*-Value
Male gender (*N* = 782)	16.9%	18.8%	1.135 (0.875–1.471)	0.341
Cancer				
- All cancer patients (*N* = 435)	24.2%	15.4%	1.739 (1.324–2.286)	<0.001
- No other obvious cause (*N* = 301)	28.5%	18.2%	1.802 (1.309–2.479)	<0.001
- On OAC treatment (*N* = 47)	20.5%	19.7%	0.834 (0.425–1.969)	0.679
DVT (*N* = 51)	88.2%	15.5%	40.91 (17.26–96.9)	<0.001
Pneumonia (*N* = 385)	15.8%	18.6%	0.853 (0.658–1.107)	0.228
AcHF				
- All patients (*N* = 207)	3.5%	20.1%	0.141 (0.065–0.302)	<0.001
- No cancer (*N* = 165)	2.5%	21.5%	0.117 (0.043–0.320)	<0.001
- On OAC treatment (*N* = 46)	2.2%	24.1%	0.089 (0.013–0.523)	0.016
- No cancer and on OAC (*N* = 37)	0%	26.1%	0.037 (0.002–0.627)	0.022
eCOPD				
- All patients (96)	5.2%	18.7%	0.238 (0.096–0.592)	0.002
- Hypercapnia (PaCO_2_ > 5.6 kPa) (*N* = 39)	0%	16.7%	0.062 (0.005–1.192)	0.054
Dyspnea				
- All patients (*N* = 850)	20.6%	17.1%	1.513 (1.158–1.976)	0.002
- No other obvious cause (*N* = 460)	27.8%	15.5%	2.098 (1.527–2.881)	<0.001
Cough (*N* = 532)	19.2%	17.2%	1.140 (0.870–1.494)	0.341
- No other obvious cause (*N* = 276)	25.7%	19.7	1.411 (1.015–1.961)	0.038
Chest pain (*N* = 339)	21,8%	16.5%	1.379 (1.025–1.863)	0.033
- No other obvious cause (*N* = 238)	23.9%	20.6%	1.212 (0.856–1.718)	0.272
Hemoptysis (*N* = 121)	16.5%	21.9%	0.902 (0.548–1.484)	0.684
- No other obvious cause (*N* = 86)	17.4%	21.8%	0.756 (0.423–1.250)	0.245
Fever (*N* = 244)	16.8%	18.1%	0.914 (0.635–1.132)	0.629
Tachycardia (>100/min)				
- All patients (*N* = 758)	14.6%	21.6%	1.479 (1.237–2.101)	<0.001
- No other obvious cause (*N* = 409)	27.6%	16.9%	1.828 (1.311–2.547)	<0.001
Hypercapnia (PaCO_2_ > 5.6 kPa)	13.3%	21.8%	0.548 (0.341–0.881)	0.013
Hypoxemia (SpO_2_ < 94%)				
- All patients (*N* = 158)	20.9%	22.7%	1.161 (0.707–1.905)	0.556
- No other obvious cause (*N* = 56)	37.5%	18.9%	1.261 (1.335–4.971)	0.005
No other obvious cause of symptoms (*N* = 599)	21.2%	16.1%	1.675 (1.278–2.195)	<0.001
- Patients aged 65 years or older (*N* = 554)	22.2%	13.6%	1.801 (1.226–2.148)	<0.001
- Patients aged 75 years or older (*N* = 288)	23.9%	14.1%	1.913 (1.248–2.932)	<0.001
OAC treatment				
- All patients (*N* = 199)	19.1%	17.7%	1.097 (0.749–1.603)	0.635
- No cancer (*N* = 152)	19.7%	14.7%	1.417 (0.915–2.193)	0.118
COVID-19 infection (*N* = 80)	21.3%	17.7%	1.254 (0.722–2.178)	0.421

AcHF—acute congestive heart failure; DVT—deep vein thrombosis; eCOPD—chronic obstructive pulmonary disease exacerbation; OAC—oral anticoagulant; other obvious causes—pneumonia, heart failure, or eCOPD established prior to CT pulmonary angiography order.

**Table 2 diagnostics-15-00827-t002:** Logistic regression for factors associated with the presence of pulmonary embolism.

Characteristic	OR (95% CI)	*p*-Value
AcHF	0.132 (0.026–0.663)	0.014
eCOPD	0.057 (0.064–0.511)	0.011
Pneumonia	0.353 (0.126–0.511)	0.048
Dyspnea	4.768 (1.390–16.357)	0.013
Hypercapnia (PaCO_2_ > 5.6 kPa)	1.406 (0.456–4.338)	0.552
Hypoxemia (SpO_2_ < 94%)	1.725 (0.642–4.637)	0.279
Pain	0.502 (0.150–1.680)	0.264
Cancer	1.289 (0.462–3.599)	0.6326
D-dimer	1.109 (1.948–1.173)	<0.001
Pulse rate	1.026 (1.008–1.044)	<0.001

AcHF—acute congestive heart failure; eCOPD—chronic obstructive pulmonary disease exacerbation.

**Table 3 diagnostics-15-00827-t003:** Comparison of ROC curve analyses for D-dimer diagnostic value in patients in whom diagnosis potentially explaining symptoms was established prior to CT pulmonary angiography.

Characteristic	AUC (95%CI)	D-Dimer Cut-Off	Sensitivity	Specificity
Whole population	0.768 (0.743–0.792)	0.59 mg/L	100%	2.7%
AcHF	0.900 (0.843–0.942)	3.68 mg/L	100%	56.6%
Pneumonia	0.771 (0.717–0.818)	1.34 mg/L	100%	24.3%
eCOPD	0.741 (0.635–0.829)	1.26 mg/L	100%	26.8%
Any other cause identified	0.793 (0.753–0.829)	1.28 mg/L	100%	22.9%

AcHF—acute congestive heart failure; eCOPD—chronic obstructive pulmonary disease exacerbation.

## Data Availability

The data presented in this study are available upon request from the corresponding author. The data are not publicly available due to privacy reasons.

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
