# Peer review of "Factors Predicting CT Pulmonary Angiography Results in the Emergency Department"

_diagnostics, 2025, doi:10.3390/diagnostics15070827_

Round 1
Reviewer 1 Report
Comments and Suggestions for Authors
The authors aimed to investigate factors associated with pulmonary embolism on CTPA, aiming to reduce unnecessary requests via a single-center retrospective study, analyzing 1555 CT reports from 2023. This is a timely study especially with increasing scans ordered in the ER. Patients’ variables that were evaluated highest Or (40.9) for PE was found for patients diagnosed with DVT prior to CTPA and lowest odds ratios of having PE were found for patients with acute congestive heart failure (OR = 0.141), especially in the absence of cancer (OR = 0.089) and for patients with hypercapnia in COPD exacerbation (OR = 0.062). Tachycardia and hypoxemia were the physiological variables positively associated with PE while hypercapnia was negatively associated with PE. For patients with heart failure, COPD exacerbation and pneumonia, higher D-dimer cut-off values (3.87 mg/L, 1.25 mg/L and 1.34 mg/L respectively) were found to retain 100% sensitivity for PE. The study is well conducted with only limitation being retrospective nature of study. I don't have any changes to recommend.
Author Response
This reviewer had no objections.
Reviewer 2 Report
Comments and Suggestions for Authors
Content suggestions:
- Can the Authors specify the family history of the included patients ?
- I would like to kindly ask the Authors whether they have any information about the antithrombin level of the studied population and whether it was substituted with the awaited effectiveness.
Author Response
Comment 1: "Can the Authors specify the family history of the included patients ?"
Response to comment 1: We are very grateful for this comment.
Family history is not routinely documented in cases of suspicion of pulmonary embolism in our emergency department, since it is not in any of the widely used pre-test probability scoring systems. Therefore, the retrospective nature of the study and sourcing the data from medical documentation precluded the collection of this information.
Given the fact that some research show that family history may be a risk indicator for pulmonary embolism (JAMA 2009) we have added a comment on this issue in the Discussion (lines 327-332 of the revised manuscript) including the mentioned reference.
Comment 2. I would like to kindly ask the Authors whether they have any information about the antithrombin level of the studied population and whether it was substituted with the awaited effectiveness.
Response to comment 2: In our hospital, antithrombin levels are not assessed in the emergency department. Therefore we do not have the data.
Reviewer 3 Report
Comments and Suggestions for Authors
Thank you for the opportunity to review this interesting and well written manuscript aimed at elucidating factors predicting CT pulmonary angiography results in the emergency department. Authors in this retrospective study showed that applying more strict criteria for CTPA orders when other pulmonary pathologies may explain respiratory symptoms could improve CTPA yield, as well as wavering CTPA orders for diagnosing PE in cases of DVT with concurrent symptoms suggestive of PE may be a choice, CTPA should however be strongly advised in patients with respiratory symptoms coupled with hypoxemia and/or tachycardia that cannot be explained with first-line diagnostic methods. Of course, all concluding remarks are derived in the context of a single-center setting with absence of preset criteria for PE on CTPA, as mentioned in limitations of the study. Nevertheless, real-world studies can provide complementary data on diagnostic effectiveness beyond highly-selective RCT patient populations and bridge this “efficacy-effectiveness” gap, providing valuable sources of real-world data.
Some minor comments follow:
Further enrichment of the introduction is suggested by clearly mentioning diagnostic approaches for PE based on current guidelines.
Did authors calculate GENEVA clinical prediction rule for pulmonary embolism in enrolled patients? Could they provide us these data?
In discussion section, authors should also mention the possibility of including selected comorbidities- beyond cancer- in the assessment of clinical (pre-test) probability for PE, following further research. This should be the main message that makes this study important. Authors should comment in more detail on this.
Author Response
Comment 1: Further enrichment of the introduction is suggested by clearly mentioning diagnostic approaches for PE based on current guidelines.
Response to comment 1: We are grateful to the reviewer for this suggestion
We have added a paragraph in the Introduction section of the revised manuscript (lines 56-62). The approach used in our hospital was mentioned in the Methods section (line 93).
Comment 2: Did authors calculate GENEVA clinical prediction rule for pulmonary embolism in enrolled patients? Could they provide us these data?
Response to comment 2: Thank you for this comment. We do not use Geneva prediction rule so it was not included in the evaluation. We could retrospectively calculate Geneva score for our patients but since it was not used in the decision making for our patients and the aim and purpose of this investigation was not comparing the scores, we feel it is better not to include it.
Comment 3: In discussion section, authors should also mention the possibility of including selected comorbidities- beyond cancer- in the assessment of clinical (pre-test) probability for PE, following further research. This should be the main message that makes this study important. Authors should comment in more detail on this.
Response to comment 3: Again thank you for this valuable suggestion.
We have included a paragraph in the Discussion section (lines 294-298), addressing this issue.